# Compressed Sensing Data with Performing Audio Signal Reconstruction for the Intelligent Classification of Chronic Respiratory Diseases

**DOI:** 10.3390/s23031439

**Published:** 2023-01-28

**Authors:** Timothy Albiges, Zoheir Sabeur, Banafshe Arbab-Zavar

**Affiliations:** Department of Computing and Informatics, Bournemouth University, Bournemouth BH12 5BB, UK

**Keywords:** artificial intelligence, machine learning, COPD, compressed sensing, signals reconstruction, dictionary learning

## Abstract

Chronic obstructive pulmonary disease (COPD) concerns the serious decline of human lung functions. These have emerged as one of the most concerning health conditions over the last two decades, after cancer around the world. The early diagnosis of COPD, particularly of lung function degradation, together with monitoring the condition by physicians, and predicting the likelihood of exacerbation events in individual patients, remains an important challenge to overcome. The requirements for achieving scalable deployments of data-driven methods using artificial intelligence for meeting such a challenge in modern COPD healthcare have become of paramount and critical importance. In this study, we have established the experimental foundations for acquiring and indeed generating biomedical observation data, for good performance signal analysis and machine learning that will lead us to the intelligent diagnosis and monitoring of COPD conditions for individual patients. Further, we investigated on the multi-resolution analysis and compression of lung audio signals, while we performed their machine classification under two distinct experiments. These respectively refer to conditions involving (1) “Healthy” or “COPD” and (2) “Healthy”, “COPD”, or “Pneumonia” classes. Signal reconstruction with the extracted features for machine learning and testing was also performed for securing the integrity of the original audio recordings. These showed high levels of accuracy together with the performances of the selected machine learning-based classifiers using diverse metrics. Our study shows promising levels of accuracy in classifying Healthy and COPD and also Healthy, COPD, and Pneumonia conditions. Further work in this study will be imminently extended to new experiments using multi-modal sensing hardware and data fusion techniques for the development of the next generation diagnosis systems for COPD healthcare of the future.

## 1. Introduction

The World Health Organization (WHO) reported that chronic obstruction pulmonary disease (COPD) was the fifth leading cause of death in the world at the beginning of the century [1]. However, in 2018, ref. [2] reported that COPD was the third largest cause of mortality in the world, and now, ref. [3] expects COPD deaths to grow to the leading cause of death by 2030. COPD is a complex respiratory disease defined as a degenerative inflammatory condition that chronically limits airflow for many pulmonary disorders [4]. 

Patients with COPD have acute exacerbations that may lead to emergency hospitalization; however, they are more likely to be re-hospitalized after their initial discharge [5]. The cost of healthcare for COPD is substantial, and expectations are that the costs will grow even more as COPD prevalence increases [2]. In the U.K. alone, ref. [6] reported that the cost of COPD reached £1.9 billion a year to the National Health Service (NHS). Hence, the prevention, early detection, and management of COPD conditions is an essential strategy for health care services [7]. Therefore, there is a need for the advancement of new decision support systems, which enable clinicians in monitoring, intelligently detecting, and understanding COPD conditions, leading to early preventions of likely exacerbation events. These systems will also serve the rest of the clinical community into pursuing their specific care operations more efficiently, including the timely drug delivery for COPD patients in their homes. Currently, the amount of effort for monitoring patients with COPD, both in their homes and hospitals, requires large deployments of medical care staff, which has become unsustainable. It is, therefore, important to opt for other approaches to meet the care needs of COPD patients of the present time and future. With the advancement and affordability of wearable sensors, information, and communication technologies, over the last two decades, it has become possible to generate large observation and measure big data, which can be efficiently analyzed for critical health conditions and processes, understanding, and extraction of knowledge. For the case of chronic respiratory diseases, there is a potential of exploring sensors and measurements-based patients signal data nowadays for the real-time analysis and diagnosis of their health conditions and sub-conditions potentially. The confirmation of these conditions using machine learning and classification methods may lead us to understand the likelihood of critical exacerbation events with lung function failures, which may occur for COPD patients and others with similar respiratory conditions.

In particular, adventitious lung sounds may occur on top of healthy lung sounds due to damage or obstructions of the lungs and airways. When observed and measured, they are normally classified into two main categories: Continuous, around 250 ms and discontinuous, about 25 ms [8]. Continuous lung sounds, such as wheezing, are often heard in conditions, such as chronic obstruction pulmonary disease (COPD), and discontinuous sounds, such as crackles, are common in pneumonia [9]. Additionally, within the respiratory auscultations, the background noise of the heart, digestive system, and internal and external noise can be heard, causing a low signal-to-noise ratio. Plus, the sounds overlap in the time and frequency domains, while the breathing rhythm makes the signals non-stationary by nature, with changing statistics over time.

As a result, the reconstruction and classification of respiratory auscultation present challenges from non-stationary signals, transient signals of discontinuous crackle sounds, to noises that can all overlap in time and frequency space. Notwithstanding, auscultation recordings are generated from a single sensory point, while the sounds are from multiple locations of the three-dimensional lung organ. This indeed creates multiple challenges in separating these mixed sounds [10].

A common theme will be in using a low band or pass filter to separate heart sounds from lung sounds [9,10,11]. However, low pass filters can induce unwanted artifacts or aliasing [12], that is undesirable noise. Additionally, ref. [11] found that not cutting out the heart sounds had negligible effects on the results. Therefore, separating the heart sounds may not be an essential step here.

Researchers have used a range of transform methods for time-frequency analysis for adventitious lung sounds and lung conditions from short-time Fourier transforms (STFT) [11,12,13], empirical mode decomposition (EMD) [13,14] to wavelet transforms (W.T.) [10,11]. Noticing the STFT and EMD have challenges in extracting features of non-stationary, transient, and overlapping signals where [11] suggests that Fourier-based methods cannot detect transient signals. Plus, ref. [14] shows that EMD can detect crackles, however, cannot distinguish overlapping lung sound crackles; this is in line with [15], where EMD works more effectively on non-overlapping signals. In addition, according to [11], the technique of utilizing continuous wavelet transform and STFT was an improvement on STFT alone. Through multi-resolution analysis, W.T. can capture more delicate signal details [12]. In addition, multi-resolution analysis in structural health monitoring of mechanical equipment has shown the ability to identify impulse and transient signals within noise [16]. Therefore, the W.T. form of multi-resolution offers a range of capabilities that aids in extracting features from respiratory auscultation audio over STFT and EMD.

The STFT, EMD, and W.T. all have inverse transforms; however, there is little research on signal reconstruction of respiratory auscultations from important representative features. Although, ref. [13] utilized compressed sensing and signal reconstruction to transmit the respiratory auscultation audio from a sensor to a smartphone. An essential factor in rebuilding the signal can map the output feature back to the input and shows the features selected capture the most important information in the original audio signals. Therefore, signal reconstruction is an essential part of this research work before we utilize most of its dominant features for respiratory diseases classification using machine learning. This paper is, therefore, purposely set out and presented in the following way: The data used in the study, the data cleaning process, the data transformation and feature reduction methods, and the reconstruction results. We then proceed with a review and implementation of classification methods and major results, leading to summarizing our findings, a discussion, and a conclusion with recommended future work.

## 2. Materials and Methods

The data utilized in this study was the ICBHI Respiratory challenge database [17]. The dataset contains 920 audio recordings of 126 patients. The audio samples vary in the number of channels (Mono and Stereo), sampling rate (4000–44,100 Hz), and duration (30–90 s). There is accompanying information on patient diagnosis and demographics for each patient. For this study, we used the Healthy, COPD, and Pneumonia of diagnosis classes of auscultation. Table 1 shows the classes used, breaking down demographics per class.

As modelling requires the data samples to be of the same length and the audio samples varied in duration, a random seven-second section was selected, which could capture a breathing cycle, where a breathing cycle ranges from 12–18 revolutions per minute [18]. Because of the imbalances between classes, the Healthy and Pneumonia classes of audio sections had two data augmentation options, out of five, applied to ensure each sample was different from each other. The augmentation options are time-stretching [19,20], where audio is sped up or down; pitch-shifting [20,21], where the audio frequency is moved up or down; added noise [19], where extra noise is added; time-shifting [20], where time is rolled forward or backward; and no augmentation. Two out of five options gave permutations of up to 20 different options, allowing for each sample to be augmented differently. The process increased the Healthy class from 35 to 735 audio samples and the Pneumonia class from 39 to 740 audio samples.

### 2.1. Audio Cleaning and Normalization

The pre-processing cleaning stage reduces noise and places all samples into a normalized format. The process contains the following steps:Thresholding;Signal smoothing;Detrending;Audio loudness normalization;Normalization.

When loading the audio, the audio samples are down sampled to 4000 Hz, bringing all samples into the same sample rate. Outliers in the audio amplitude, expected by stethoscope contact movement, were reduced by thresholding. By thresholding the signal amplitude above four standard deviations and reduced to the mean, crackles can appear within four standard deviations. With down-sampling and removing outliers, cleaning the audio with a smoothing filter will also remove some noise. The choice of filter is the Savol filter, a moving filter with a polynomial function that is well suited for noise reduction for lung sounds [22]. The audio samples are non-stationary and can display trending; therefore, detrending reduces the non-stationary [23] (p. 47). The works of [24] highlight that respiratory audio has two components: air turbulence and lung structural sounds, which compete with each other when listened to from different locations. Therefore, the EUB R128 normalization is used. Finally, the values are normalized to bring them into the same range.

### 2.2. Wavelet Transform

Wavelet transform (W.T.) is used for multi-resolution audio signals, breaking them down into different levels of frequency ranges, where the formula is shown in Equation (1). The mother wavelet (**Ψ***) chosen is the Morlet wavelet because the distribution characteristics are similar to the transient crackle with a sudden peak.
(1)$$w_{n}\left(s\right)=\overset{N-1}{\underset{n^{\prime }=0}{\sum }}x_{n^{\prime }}\Psi^{*}\left[\frac{\left(n^{\prime }-n\right)\delta t}{s}\right]\;$$

The complex Morlet wavelet returns the real and imaginary components that this study will analyze. This analysis supports our objectives as W.T. is robust to noise, localizes audio characteristics [12], and has inverse transform [25]. The inverse transform allows for the reconstruction of the signal from the multi-resolution analysis to audio signals

### 2.3. Compressed Sensing

Compressed sensing underlines the sparse encoder dictionary learning. The main principles of compressed sensing are:Incoherence;Sparsity.

Incoherence is a property in that the samples are not connected by time or spatial domains, which expanses the time-frequency localization problem or uncertainty problem. In that, the samples are more spread out and sparse within the domain [26]. Whereas in compressed sensing matrices, the values in the rows do not correlate with those in the columns [27] (p. 90). Sparsity is a property where samples are spread out, where the low values nearing zero can be zeroed out altogether. This allows the data to have minimal or low non-zero elements. The sparsity constraint placed on compressed sensing enables the change from an over-complete solution to be relaxed and a unique solution to be found [28]. When compressed sensing comes to matrix forms, the matrix structure, which maps linearly when restricted to sparsity [29] naturally preserving the so-called restricted isometric property (RIP) [27] (pp. 90–96). The ability to sub-sample from subspace aids feature reduction, which is with less than the Nyquist sample rate, which allows for meeting the objective of signal reconstruction.

### 2.4. Dictionary Learning

Dictionary leaning incorporates compressed sensing with the factors of sparsity by relaxing the linear constraints and utilizing an error-bound element and incoherence factor between each atom (column) in the dictionary [28]. Additionally, dictionary learning uses algorithms, such as gradient descent or orthogonal matching pursuit (OMP) to aid in finding a sparse representation and reconstruction process by selecting highly correlated samples for the dictionary atom [30]. Dictionary learning is calculated by Equation (2).
(2)$$\left[u,v\right]=argmin\frac{1}{2}{\left|\left|X-UV\right|\right|}_{L2}+\alpha \times {\left|\left|U\right|\right|}_{L1}\left(U,V\right)\;\mathrm{with}\;{\left|\left|V\_k\right|\right|}_{2}<=1\;for\;all\;0<=k<n\_components\;$$

Dictionary learning supports the decomposition of the multi-resolution analysis matrix into a reduced number of components; the multiplication of the components and the transform results in reconstructing an approximation of the multi-resolution analysis matrix.

### 2.5. Singular Value Decomposition

Singular value decomposition (SVD) is a method that factorizes real, or complex, matrices into three matrices. It is often used in signals processing in order to compress signal data to their most representative matrix form of features and make it more efficient to work with complex signals. Specifically, the method exposes many of the important and interesting representational features of signals from the original matrix. For an illustration and the special case of real matrices, SVD is performed as follows: (3)$$A=U\sum V^{T}\;$$
where ***A*** is a (*n* × *p*) matrix to decompose [31]. U is a (*n* × *n*) orthogonal matrix, whose columns are known as the left-singular vectors; ∑ has the same dimensions (*n* × *p*) as ***A*** and has the so-called singular values in its diagonal. $V^{T}$ is an orthogonal (*p* × *p*) matrix, which is the transpose matrix of V, whose rows are known as the right singular vectors. Further, SVD computations involve the extraction of the eigenvalues and eigenvectors of ***AA^T^*** and ***A^T^A***. Their eigenvectors make up the columns of V and U, respectively. The singular values are the diagonal elements of the ∑ matrix. They are usually arranged in descending orders. Additionally, they are the square roots of the eigenvalues of ***AA^T^*** or ***A^T^A***, ref. [32]. In addition, we note that SVD supports signals of noise reduction, in this case, through matrix characteristics decomposition, which leads to the most interesting number of features representing the signal, while assuring the ability to recover the original matrix through SVD matrices operations. 

### 2.6. Signal Reconstruction Metrics

In order to understand the accuracy of the signal reconstruction, comparing the pre-processed signal with the reconstructed signal will highlight the accuracy. Therefore, the mean square error (MSE) and the correlation coefficients can be used as metrics for signal similarity analyses.

The mean square error is a measure of the difference calculated by Equation (4) [30], where *A* is the original signal and *B* is the reconstructed signal. The MSE shows the average difference in the distance between two signals.
(4)$$MSE=\frac{\sum_{n}{\left(A\left[n\right]-B\left[n\right]\right)}^{2}}{m}$$

Another measure of signal similarity is the correlation coefficient between the two signals **A** and **B** [33], as calculated by Equation (5).
(5)$$Corr\;Coef=\frac{\sum^{\;}\left(A_{i}-\bar{A}\;\right)\sum^{\;}\left(B_{i}-\bar{B}\right)}{\sqrt{\sum^{\;}{\left(A_{i}-\bar{A}\right)}^{2\;}\sum^{\;}{\left(B_{i}-\bar{B}\right)}^{2\;}}}$$
where $\bar{A}$ is the original signal mean, and $\bar{B}$ is the recovered signal mean. The correlation coefficient shows the linear dependence between the signals. 

#### 2.6.1. Summary of Extracted Features

The framework extracted features is where U contained 153 features, *V^T^* contained 90 features, and S contained 9 features. The number of features was the same for the real and imaginary components of the signals.

#### 2.6.2. Signal Reconstruction Results

The results of signal reconstruction are shown in Table 2 below.

#### 2.6.3. Summary of Signal Reconstruction

The results of the MSE show the reconstruction accuracy averages at 3 × 10^−3^, with the best result reaching 5 × 10^−4^, meaning that the distance between the pre-processed and reconstructed signals is very small. Likewise, the correlation coefficients have a mean score of 0.57, while the highest score reaches 0.92. Reconstruction results demonstrate that the reconstruction is an excellent approximation of the pre-processed original audio signal.

### 2.7. Classification

The study covered two different classifications, one of “Healthy” and “COPD” and the second of “Healthy”,” COPD”, or “Pneumonia”. Pneumonia was chosen as the adventitious sounds are mainly crackles, whereas COPD is mainly wheezing, which allows for discrimination between the two classes. As the complex Morlet wavelet gives the real and the imaginary components of the signal, each component is classified. The models for classification are: The Gaussian mixture model (GMM), decision tree classifier (DTC), support vector machine (SVM), and random forest classifier (RFC).

The GMM is a classification algorithm, which allows for overlapping borders of Gaussian distribution clusters that may support the overlapping frequencies of lung sounds [34]. DTC uses a divide-and-conquer strategy for classification that offers transparency and, therefore, allows for an objective analysis [35]. The SVM utilizes a boundary separation, or if data are highly dimensional, a separation of categories with a hyper-plane, which can be linear, polynomial, quadratic, or of higher orders [35]. The RFC is an ensemble approach, a powerful tool for data mining in which the combining of multiple trees for the outcome can be viewed as a bias-variance decomposition. Specifically, it aids the performance [35], which is supported by the random bagging of sampling with replacement from the training data and bootstrap of the features [36]. Additionally, random forests can give information on feature importance; therefore, it is an excellent option for classification. Grid search, which cycles through different parameters for the models to find the optimal parameters, is used to increase the model’s performance. The grid search parameters for the RFC number of estimators range from one hundred to six hundred with increments of fifty, and the depth range from ten to one hundred with increments of ten. 

#### Classification Metrics

The performance of the models is evaluated by looking at the true positives (T.P.), true negatives (T.N.), false positives (F.P.), and false negatives (F.N.) [10]. We utilized the accuracy, F1 scores, receiver operator characteristic (ROC) curves, and area under curve (AUC) scores [36]. For the Healthy, COPD, and Pneumonia classifications, the ROC curves will be the one-versus-all classification, which compares one class to the other two classes. five-fold cross-validation is utilized, while the results are the averages across the five-fold and the cross-validation standard deviation to ensure that the model performance is robustly assessed. The level of coverage of the model’s performance is reported with confidence intervals of 95% [36].

## 3. Results

### 3.1. Healthy and COPD Classification Results

The results are set out with baseline results, model parameter optimization results, the ROC, and the area under the curve plots. The baseline results for the classification of healthy and COPD is shown in Table 3.

Taking the SVD and random forest further with parameter tuning, the results are shown in Table 4. Cross-validation scores and confidence intervals are reported.

ROC curves are used to display the discriminative ability of the classification models. The comparison of the different models are shown in Figure 1, and the comparison of the real and imaginary components using Random forest classifier ROC curve results are shown in Figure 2.

The ROC curve results for the classification of Healthy, COPD, and pneumonia are shown in Figure 3 and Figure 4 below.

### 3.2. Healthy, COPD, and Pneumonia Classification Results

The baseline results for the classification of healthy, COPD, and pneumonia are shown in Table 5.

The random forest and SVC classifiers were the best performing and taken forward for parameter tuning; the results are shown in Table 6.

### 3.3. Summary of Classification Findings

The random forest models produced the best performing models for the classification of Healthy versus COPD and Healthy versus COPD versus Pneumonia. The best features for the Healthy versus COPD classification were the SVD’s U and *V^T^* for the imaginary component of the auscultation’s audio, both having accuracies of 80% and the area under ROC curves showed that the SVD U elements were better at discriminating between healthy and COPD than the SVD *V^T^* elements with values of 0.87 and 0.77, respectively, with the random forest model. Similarly, for the classification of Healthy versus COPD versus Pneumonia, the best results were from the random forest classifier, highlighted in Figure 3. However, the best features were on the SVD’s S (Singular) values of both the real and imaginary components of the auscultation recordings, while achieving 70% and 68% accuracy, respectively. The random forest model’s ability to discriminate between classes on the SVD S elements was relatively close values, with the real components ranging between 0.82 to 0.86 (see Figure 3c) and the imaginary components ranging from 0.80 to 0.83 (see Figure 3f).

## 4. Discussion

There are some encouraging results in the classification of Healthy and COPD; the imaginary components of the signal and the orthogonal SVD elements are the best performers, which may relate to the harmonic resonance of wheezes often identified in COPD patients. The classification of the Healthy versus COPD achieved a good accuracy of 80%, with 95% confidence levels of 76–79% on the audio signals imaginary components on the SVD’s U and V.T. elements. For the Healthy versus COPD versus Pneumonia, an acceptable level of accuracy of 70% with a 95% confidence level of 66–70% on the audio signals real components on the SVD’s S (singular values), with good levels of discrimination between conditions. For the signal reconstruction, the best scores are MSE of 5.2 × 10^−3^ with a mean score of 3.0 × 10^−2^ and a correlation coefficient score of 0.92 with a mean score of 0.57. Indeed, this suggests a good level of signal recovery. When comparing the results in the Healthy versus COPD versus Pneumonia, we find that the best performance was from the real component of the signal with the SVD’s element, which relates to the signal’s strength, especially between the COPD and Pneumonia that had higher classification numbers in the confusion matrix. 

In comparison, ref. [11], who also utilized W.T., achieved scores of 39.97–49.86% in classifying normal lung sounds, wheezes, and crackles on the ICBHI 2017 challenge database. Ref. [11]’s choice of adventitious sounds can be related to Healthy, COPD, and Pneumonia, respectively, in which this study demonstrated higher accuracies of classification. In addition, ref. [37] discusses the challenge of achieving above 50% accuracy in the ICBHI 2017 challenge database, where they aimed to classify normal, wheezes, crackles, and both wheezes and crackles. Ref. [37] suggested that there may be issues with the dataset as they found an audio of a patient diagnosed with respiratory disease, but the annotated notes for the specific audio recording had no adventitious sound noted. However, no adventitious sounds do not mean a lack of disease, as [38] noted. Ref. [39] utilized discrete wavelet transforms and deep learning for classifying the ICBHI 2017 challenge database into healthy and unhealthy, which achieved an F1 score of 81.64%, similar to the F1 scores of the best models of Healthy versus COPD of 83%. However, this study’s approach was more focused on COPD, whereas [39] unhealthy had a broader range of diseases. Ref. [40] achieved high accuracy of 92.30% by utilizing a 17-layered 2D-convolutional neural network (CNN) with features of MFCC and spectrograms to classify the ICHBI dataset auscultation recordings into their associated diseases. 

The advantage of our proposed approach is the ability to achieve signal reconstruction and recovery to approximate the original signal with high credibility. Furthermore, the recovery of our signals to their high level of accuracy, together with the good levels of their correct classification rates on the health conditions using machine learning highlights, is a way forward for understanding human respiratory conditions. Our method is specifically feasible for respiratory auscultation classifications and supports the hypotheses on health conditions.

In addition, while other work has focused on statistical and neural network-based approaches, our results demonstrate a new method of utilizing compressed sensing for auscultation classifications. Nevertheless, further optimization of the extraction process needs to be deployed together with large volumes of experimental datasets to increase the accuracy of both signal recovery and machine classifications. In future work, experimenting with multi-modal data and dictionary learning for improving the diagnostic and prognosis of COPD conditions should be the focus.

## 5. Conclusions

The developed benchmark work in this study not only provides good levels of accuracy for signal reconstruction, but it also brings good performing machine classification of respiratory lung sounds. These are brought in good context of their associated chronic health conditions. Specifically, on the machine classification side, the random forest classifier is the performing algorithm with accuracies ranging from around 80% for classifying cases of “Healthy” and “COPD”. It reaches accuracies of approximately 70% for classifying cases, including “Healthy”, “COPD”, and “Pneumonia”. These were all obtained with confidence intervals showing the stability of the models. The ROC curves show the discrimination ability of the classifiers, although with limitations. Our work has also the potential of applications in other respiratory disease classifications and beyond. However, more work needs to be performed, since we need to improve the performance of our classifiers to higher levels first while validating them under much larger and diverse datasets. Our future work will specifically involve research investigations on obstructive pulmonary chronic respiratory diseases using larger datasets in order to scale our approaches in terms of their accuracies and performances. We will also aim to identify lung sounds that correspond to various sub-conditions of COPD, particularly those which may highly lead to patients’ exacerbation events. We will aim in the near future to automatically predict the likelihood of occurrence of such serious events, ahead of time and with good contexts, in order to accelerate medical responses to patients under critical respiratory conditions. 

## Figures and Tables

**Figure 1 sensors-23-01439-f001:**
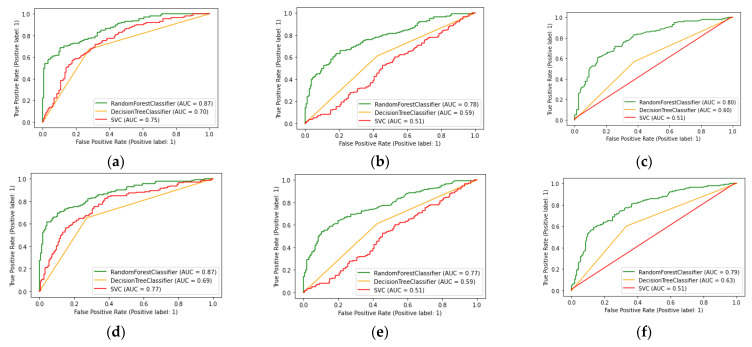
ROC curves of classification models of each SVD element and real and imaginary components: (**a**) ROC curves of real components of SVD U element; (**b**) ROC curves of real component of SVD *V^T^* element; (**c**) ROC curves of real component of SVD S element; (**d**) ROC curves of imaginary component of SVD U element; (**e**) ROC curves of imaginary component of SVD *V^T^* element; (**f**) ROC curves of Imaginary component of SVD S element.

**Figure 2 sensors-23-01439-f002:**
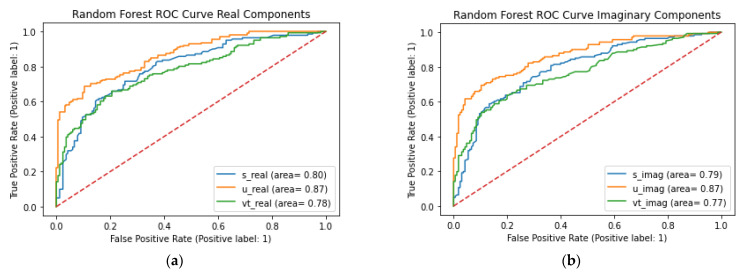
ROC curve of classification of healthy and COPD classifications of each signal component and SVD elements: (**a**) Chart of the ROC curve of the RFC for real components on the classification of Healthy and COPD in the first panel; (**b**) chart of the ROC curve of the RFC for imaginary components on the classification of Healthy and COPD.

**Figure 3 sensors-23-01439-f003:**
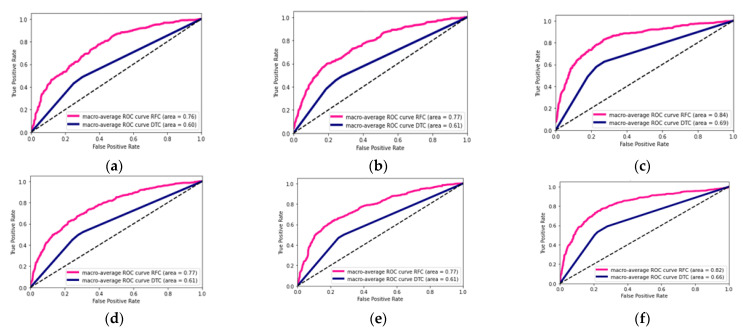
DTC and RFC classification of Healthy, COPD, and Pneumonia for each SVD element and real and imaginary components of the signals: (**a**) ROC curves of real components of SVD U element; (**b**) ROC curves of real component of SVD *V^T^* element; (**c**) ROC curves of real component of SVD S element; (**d**) ROC curves of imaginary component of SVD U element; (**e**) ROC curves of imaginary component of SVD *V^T^* element; (**f**) ROC curves of imaginary component of SVD S element.

**Figure 4 sensors-23-01439-f004:**
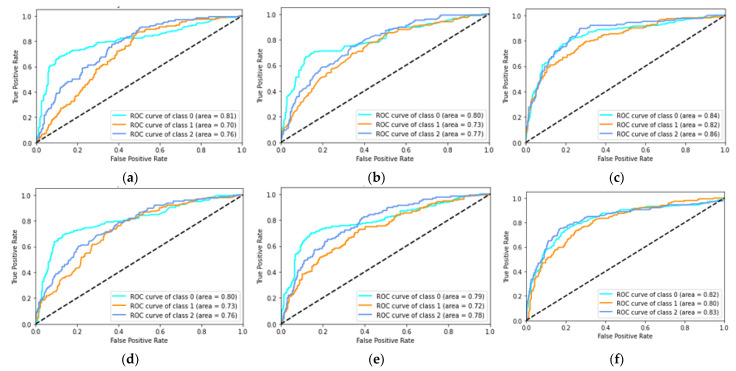
RFC classification of Healthy, COPD, and Pneumonia one-vs-rest ROC curves for each SVD element and real and imaginary components of the signals: (**a**) ROC curves of real components of SVD U element; (**b**) ROC curves of real component of SVD *V^T^* element; (**c**) ROC curves of real component of SVD S element; (**d**) ROC curves of imaginary component of SVD U element; (**e**) ROC curves of imaginary component of SVD *V^T^* element; (**f**) ROC curves of imaginary component of SVD S element.

**Table 1 sensors-23-01439-t001:** ICBHI 2017 challenge database selected class breakdown.

Conditions	Number of Recordings	Biological Sex (Count)		Age Range (Years)	
		Male	Female	Min	Max
COPD	793	512	266	45	93
Healthy	35	15	20	0.25	16
Pneumonia	37	30	7	4	81

**Table 2 sensors-23-01439-t002:** Result summary of signal reconstruction from extracted features.

Stats	MSE	Correlation Coefficient
count	2268	2268
mean	0.030668	0.576079
std	0.012137	0.150377
min	0.005188	0.014053
0.25	0.022598	0.488954
0.5	0.029151	0.582712
0.75	0.036262	0.682031
max	0.142799	0.924803

**Table 3 sensors-23-01439-t003:** Healthy vs COPD classification baseline results. All the baseline results have been achieved with the following parameter settings: Random forest (RFC): d = 500, e = 280 (in these, *d* stands for depth, and *e* stands for the number of estimators); GMM: components = 2, covariance = full; SVC: gamma = auto, C = 3000.

Classification Details	Classification Model	F1-Score	Accuracy
SVD U, Real	RFC, d = 500, e = 280	78.5	80
	GMM, components = 2	33.5	44
	DTC	69.5	70
	SVC, C = 3000	68.5	69
SVD Vt, Real	RFC, d = 500, e = 280	71	72
	GMM, components = 2	35	47
	DTC	59	59
	SVC, C = 3000	53.5	54
SVD S, Real	RFC, d = 500, e = 280	71	71
	GMM, components = 2	35.5	38
	DTC	60	60
	SVC, C = 3000	35	54
SVD U, Imag	RFC, d = 500, e = 280	78.5	79
	GMM, components = 2	37	53
	DTC	69	69
	SVC, C = 3000	70	70
SVD Vt, Imag	RFC, d = 500, e = 280	71	72
	GMM, components = 2	47.5	48
	DTC	59	59
	SVC	53.5	54
SVD S Imag	RFC, d = 500, e = 280	71	72
	GMM, components = 2	35	54
	DTC	63	64
	SVC, C = 3000	35	54

**Table 4 sensors-23-01439-t004:** Healthy vs. COPD classification of parameter tuning results. In these, *d* stands for depth, and *e* stands for the number of estimators.

Classification Details	Classification Model	Macro F1-Score	Accuracy	CV Score	CV Std	CI 95%
SVD U, Real	RFC, d = 25, e = 390	78.5	79	76	5	73–78
	SVC, C = 2265.8	68.5	69			
SVD Vt, Real	RFC, d = 20, e = 400	72.5	73	68	5	65–70
	SVC, C = 17,911.6	53.5	54			
SVD S, Real	RFC, d = 25, e = 390	72	72	73	6	70–75
	SVC, C = 1251.9	35	54			
SVD U, Imag	RFC, d = 30, e = 390	79.5	80	76	5	74–79
	SVC, C = 80,190.1	70	70			
SVD Vt, Imag	RFC, d = 20, e = 400	79.5	80	76	5	74–79
	SVC, C = 58,523.6	70	70			
SVD S Imag	RFC, d = 30, e = 400	71	72	73	5	70–74
	SVC, C = 2764.8	35	54			

**Table 5 sensors-23-01439-t005:** Healthy vs COPD vs Pneumonia baseline classification results. All the baseline results have been achieved with the following parameter settings: Random forest (RFC): d = 500, e = 280 (in these, *d* stands for depth, and *e* stands for the number of estimators); GMM: components = 2, covariance = full; SVC: gamma = auto, C = 3000.

Details	Classification Model	Macro F1-Score	Accuracy
SVD U, Real	RFC, d = 500, e = 280	59.7	51
	GMM, components = 2	30.3	37
	DTC	50.7	60
	SVC, C = 3000	45	46
SVD Vt, Real	RFC, d = 500, e = 280	59.3	60
	GMM, components = 2	31	32
	DTC	46	46
	SVC, C = 3000	44.7	45
SVD S, Real	RFC, d = 500, e = 280	69.7	70
	GMM, components = 2	22	40
	DTC	55.3	56
	SVC, C = 3000	19	39
SVD U, Imag	RFC, d = 500, e = 280	60.3	61
	GMM, components = 2	48	50
	DTC	52	52
	SVC, C = 3000	46.3	47
SVD Vt, Imag	RFC, d = 500, e = 280	62.3	62
	GMM, components = 2	32.7	32
	DTC	49	50
	SVC, C = 3000	44.3	45
SVD S Imag	RFC, d = 500, e = 280	67.3	67
	GMM, components = 2	20.7	39
	DTC	58.7	59
	SVC, C = 3000	19	39

**Table 6 sensors-23-01439-t006:** Healthy vs COPD vs Pneumonia classification of parameter tuning results. In these, *d* stands for depth, and *e* stands for the number of estimators.

Classification Details	Classification Model	Macro F1-Score	Accuracy	CV Score	CV Std	CI 95%
SVD U, Real	RFC, d = 20, e = 300	58.7	59	58	3	56–59
	SVC, C = 1143.9	43.7	45			
SVD Vt, Real	RFC, d = 40, e = 500	60.3	61	59	4	57–61
	SVC, C = 1839.8	46.7	47			
SVD S, Real	RFC, d = 20, e = 400	70	70	68	4.5	66–70
	SVC, C = 1536.9	21				
SVD U, Imag	RFC, d = 30, e = 500	60.3	61	58	4.9	56–61
	SVC, C = 1536.9	46	47			
SVD Vt, Imag	RFC, d = 20, e = 400	62.3	63	59	3.8	57–61
	SVC, C = 1536.9	49	50			
SVD S Imag	RFC, d = 20, e = 300	67.7	68	68	4.2	65–70
	SVC, C = 1536.9	19.7	39			

## Data Availability

The open access dataset is available from https://bhichallenge.med.auth.gr/ICBHI_2017_Challenge.

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
