# Peer review of "Compressed Sensing Data with Performing Audio Signal Reconstruction for the Intelligent Classification of Chronic Respiratory Diseases"

_sensors, 2023, doi:10.3390/s23031439_

Round 1

Reviewer 1 Report

The manuscript has some typos and alignment issues, so authors should check the paper completely and avoid jargon words.

The introduction part might be expanded to include difficulties that arise in the context of previous work and also how chronic respiratory diseases are classified, and what are the factors considered for their classification.

Literature review approaches must be strengthened in sensing data for respiratory diseases by including the issues in the current system and how the author proposes to overcome them same. Authors may refer the papers like

1. Energy-Aware Distributed Edge ML for mHealth Applications with Strict Latency Requirements

2. A novel PCA-firefly based XGBoost classification model for intrusion detection in networks using GPU

What are the factors that are considered for incoherence and sparsity as per the encoder dictionary learning?

What is the need of singular value decomposition and there are several equations that can be used and why authors have used only summation equation with A and V.

Graphs in Figure 1 should be properly addressed with the appropriate notations.

Experimental results are not analyzed in detail.

Conclusion should be completely revised.

Reviewer 2 Report

The main proposed approach has novelty in contribution and methodology. Revision in terms of technical details is needed before publication. Also, paper organization can be improved. In this respect, some comments are suggested to describe technical details.

1. Describe the thresholding process with more technical details. What is your aim to perform threshold as preprocess?

2. How many features did you extract to evaluate Table 2?

3. How many trees and depth length did you consider in performing random forest? (Table 3)

4. Discuss the Figure 3.a with more details. What do you find from this plot?  

5. It is suggested to discuss about the limitation of your work in the conclusion briefly  

6. Don’t use dot “.” at the end of title

7. Your proposed approach can be used widely in medical disease classification problems such as lung cancer. For example, I find a paper entitled “Detection of Lung Cancer Tumor in CT Scan Images Using Novel Combination of Super Pixel and Active Contour Algorithms”, which has relation. Cite this paper and discuss about potential applications briefly.  

Round 2

Reviewer 2 Report

Just some of the comments in the previous review round are considered in the revised version. It is better to consider following comments before publication:

1. The quality of the figure 4 is too low

2. The end of all lines should be justified

3. What is the "cross-validation average" exactly? Did you use it as performance evaluation metric? Discuss about it in a clear way

4. Did you evaluate the random forest based on different "d" and "e" values?

Author Response

See attached response document. Much appreciated. 
